# Retinal Pigment Epithelial Characteristics in Acute and Resolved Vogt-Koyanagi-Harada Disease

**DOI:** 10.3390/jcm12062368

**Published:** 2023-03-19

**Authors:** Ninan Jacob, Mudit Tyagi, Jay Chhablani, Raja Narayanan, Anup Kelgaonkar, Mukesh Jain, Sumit Randhir Singh, Niroj Kumar Sahoo

**Affiliations:** 1Anant Bajaj Retina Institute, Kode Venkatadri Chowdary Campus, L V Prasad Eye Institute, Vijayawada 521134, India; 2Anant Bajaj Retina Institute, Kallam Anji Reddy Campus, L V Prasad Eye Institute, Hyderabad 500034, India; 3UPMC Eye Center, University of Pittsburgh, Pittsburgh, PA 15213, USA; 4Anant Bajaj Retina Institute, Mithu Tulsi Chanrai Campus, L V Prasad Eye Institute, Bhubaneswar 751024, India; 5Sri Sai Lions Netralaya and Sri Sai Eye Hospital, Patna 800020, India

**Keywords:** Vogt-Koyanagi-Harada disease, VKH, retinal pigment epithelium, RPE, OCT, optical coherence tomography

## Abstract

Vogt-Koyanagi-Harada (VKH) disease is an auto-immune inflammatory disease of choroidal origin. During the acute stage, optical coherence tomography (OCT), however, may not be able to assess the entire choroid. The aims of the paper were to evaluate the role of retinal pigment epithelium (RPE) as a biomarker of inflammation in acute VKH. This was a retrospective observational study done in 55 eyes of 29 patients with acute VKH. RPE thickness, total choroidal thickness, and RPE reflectivity before and after resolution were analyzed using image J software. Correlations between several baseline and post-resolution parameters were performed, and factors affecting change in visual acuity were analyzed. A significant decrease in RPE thickness and a significant increase in RPE reflectivity were seen following resolution of the disease. Furthermore, there was a significant correlation between RPE and choroidal thickness during the acute stage of the disease. Baseline visual acuity and the presence of bacillary detachment at baseline were the only factors responsible for changes in visual acuity. We propose the utility of RPE layer as a surrogate biomarker of choroidal activity and inflammation in terms of RPE reflectivity and RPE thickness during the acute stage of VKH, especially when there is poor imaging of the choroid.

## 1. Introduction

Vogt-Koyanagi-Harada (VKH) disease is an autoimmune multisystem inflammatory disorder directed against melanocytes with ocular, neurologic, dermatologic, and skin manifestations [1,2,3]. It is usually a disease of the adult population, with women being more commonly affected [4]. It is a granulomatous posterior or pan uveitis, characterized by diffuse choroiditis, vitritis, exudative retinal detachment, subretinal precipitates, and disc oedema. Treatment for VKH typically involves high-dose corticosteroids in the acute phase, often requiring additional immunomodulatory therapy with slow taper [5]. It has been shown that prompt treatment of VKH is associated with better visual outcomes and fewer ocular and systemic complications. However, the disease is often heralded by recurrent episodes of activity resulting in retinal, retinal pigment epithelial, choroidal atrophy, and subretinal fibrosis [1,3,6]. These invariably lead to permanent vision loss. Thus, early detection of the disease is of utmost importance, especially in eyes with overlapping features with other causes of subretinal fluid.

Though the diagnosis of VKH is practically a clinical one, several imaging modalities such as the ultrasound B scan, fundus fluorescein angiography (FFA), indocyanine angiography (ICGA), and optical coherence tomography (OCT) have been described, which can be invaluable, especially in cases of recurrences. VKH being a disease originating from the choroid, there have been several reports on the assessment of disease activity based on choroidal characteristics [7]. However, during the acute phase of the disease, it may not be possible to assess the entire choroid due to increased choroidal thickness and back shadowing. We realized that this backshadowing occurred even in areas without any subretinal fluid. Thus, we speculated that a few changes might be occurring at the level of the retinal pigment epithelium (RPE) that is causing these changes. While few imaging studies have analyzed RPE, both following the resolution of the disease and in the chronic phase, describing features such as focal areas of reversible RPE thickening and RPE atrophy, there is a lacuna in the literature regarding RPE imaging characteristics during the acute phase and immediately following resolution of the disease [8,9,10]. Thus, in the current study, we aimed to assess the morphological features of the more easily scanned RPE during the acute phase of the disease and following resolution and its probable role as a biomarker for the diagnosis and assessing the severity of VKH.

## 2. Materials and Methods

The study undertaken was a retrospective, observational study. The duration of the study was from January 2016 to December 2021. Ethical clearance for the study was obtained from the Institute Ethics Committee, which adhered to the tenets of the Declaration of Helsinki. Consecutive patients with acute Vogt-Koyanagi-Harada (VKH) disease were included. All cases were diagnosed according to the revised diagnostic criteria for VKH established by the international committee [11]. Patients with co-existing retinopathy or other causes of decreased vision and eyes with poor documentation/image quality were excluded from the study. Electronic medical records were reviewed to obtain data including age, sex, best-corrected visual acuity (BCVA), duration of symptoms before presentation, and duration of disease. The resolution of the disease was determined at the discretion of the treating ophthalmologist based on the resolution of inflammatory signs such as cells in the anterior chamber and the complete resolution of subretinal fluid on OCT. Improvement in visual acuity at resolution was defined as any decrease in the logarithm of the minimum angle of resolution (logMAR) value from baseline. OCT images were captured by Triton DRI OCT (Topcon, Tokyo, Japan). A 12 mm horizontal scan across the foveal centre was selected for analysis at baseline and post-resolution of disease. Image registration was done for all follow-up imaging. All patients were treated with either oral or intravenous steroids, followed by oral steroids with gradual tapering of dose, with most patients requiring additional immunomodulatory therapy.

From the OCT images, the central macular thickness (CMT), retinal pigment epithelial (RPE), total choroidal thickness (CT), as well as RPE reflectivity, were analyzed. The analysis was done using ImageJ software by Fiji. During the acute phase of the disease, RPE reflectivity was calculated within the area of RPE below the fovea, below the detached retina, and below the areas of the attached retina (Figure 1). The oval selection tool of ImageJ was used to manually mark an elliptical region of interest within the RPE to measure the reflectivity in that area, i.e., the mean grey value [12]. RPE thickness was also calculated in the subfoveal area, below the area of the attached retina and the area of the detached retina, using the in-built calliper tool in ImageJ (Figure 2). Three adjacent locations were taken in all the areas measured, i.e., under the fovea and in the areas of attached and detached retina, for both RPE reflectivity and thickness measurements, which were then averaged. Total choroidal thickness was measured in the subfoveal area. The total choroidal thickness was measured from the outer border of the RPE to the inner border of the sclera. Measurements post-resolution of the disease were done in locations identical to the pretreatment measurements on the OCT B-scan, enabled by image registration. Morphological features that were analyzed included subretinal hyperreflective dots, choroidal hyperreflective dots, bacillary layer detachment, and RPE vacuolations.

Subretinal hyperreflective dots were defined as hyperreflective dots below the neurosensory retina and over the RPE without associated back shadowing (Figure 3B). Choroidal hyperreflective dots were defined as circumscribed dots within the choroid seen on an OCT scan having equal or higher reflectivity than the RPE band (Figure 4). Bacillary layer detachment was defined as a split of the neurosensory retina at the level of the myoid zone in the photoreceptor layer (Figure 4). RPE vacuolations were defined as hypo reflective cavities within the RPE (Figure 3A).

For analysis, both eyes were considered. Eyes were excluded from analysis in cases of poor image quality or poor documentation. The statistical analysis was performed using Microsoft Excel (Microsoft Office 2019, Redmond, Washington, United States) and SPSS (SPSS v20.0, IBM corp., Armonk, NY, USA). Snellen visual acuity was converted into logMAR units. Continuous variables were expressed as mean ± standard deviation or median with interquartile range (IQR). A two-tailed paired T-test or Wilcoxon signed-rank test was used to analyze variables pre- and post-resolution. Pearson correlation was used to analyze the relationship between various baseline and final parameters. Linear regression analysis was used to assess the factors affecting change in visual acuity. We performed a generalized estimating equation (GEE) to compensate for the effect of both eyes of the same patient on the statistical analysis. A *p*-value of less than 0.05 was considered statistically significant.

## 3. Results

The study included 55 eyes from 29 patients with acute VKH, which included 9 males and 20 females. Three eyes from three patients were excluded due to poor image quality or poor documentation. The average age was 33.76 ± 12.12 years (ranging from 9 to 57 years). The median duration of symptoms (decreased vision) prior to presentation to the clinic was 7 days (IQR 5 to 17.5 days), and the median duration till resolution of disease activity was 2.3 months (IQR 0.7 to 3.8 months). All eyes had complete resolution of subretinal fluid by the last follow-up. There was a significant improvement in the mean BCVA of the cohort after the resolution of the disease. (Table 1). The patients were followed up for a mean duration of 9.7 ± 3.2 months. In terms of OCT parameters, a significant decrease in the CMT, SFCT, and RPE thickness (subfoveal, under the detached retina, and under the attached retina) was noted after the resolution of the disease. However, an increase in RPE reflectivity was seen post-resolution of the disease (Table 1). There was no significant difference in the signal strength before and after the resolution of the disease.

A few morphological features that we noticed in our study population during the acute phase of the disease were bacillary layer detachment (25/55 eyes), subretinal hyperreflective dots/ inflammatory deposits over the RPE (55/55 eyes), choroidal hyperreflective dots (47/55 eyes), and RPE vacuolations (15/55 eyes). Out of these, while bacillary layer detachment, RPE vacuolations, and choroidal hyperreflective dots disappeared in all the patients, deposits over RPE (55/55 eyes) persisted. However, no correlation could be found between the above-mentioned RPE morphological features and the choroidal thickness.

### Correlations

On comparing various parameters, a significant correlation was seen between RPE reflectivity and RPE thickness under attached retina post-resolution, SFCT and sub-foveal RPE thickness pre-treatment and baseline BCVA and duration of symptoms. Other correlations were statistically insignificant (Table 2).

On univariable linear regression analysis, a worse baseline BCVA, a higher baseline CT, a lesser duration of symptoms, a higher baseline thickness of RPE under detached retina, the presence of bacillary layer detachment, and the presence of hyperreflective choroidal dots were associated with greater improvement in visual acuity. However, on multivariable regression analysis, only the baseline BCVA and the presence of bacillary detachment were found to be significant (Table 3).

## 4. Discussion

In our study, we found that the RPE thickness in VKH patients significantly decreased while reflectivity significantly increased after resolution when compared with the active stage of the disease. This change was seen under the fovea, in areas of attached as well as detached retina. Furthermore, the thickness of the RPE correlated with the total choroidal thickness in the active phase of the disease. Moreover, the presence of bacillary layer detachment and baseline BCVA had a significant influence on the degree of visual acuity improvement.

It has been shown that the choroidal thickness in patients with active VKH is thicker as compared with the normal population and that the thickness decreases following the resolution of the disease [7,13]. Histopathological studies have shown the presence of polymorphonuclear neutrophils and macrophages in the thickened choroid, associated with oedema and vascular dilation, which explains the marked choroidal thickening during the active stage of the disease [13]. Thus, evaluation of the choroid is invaluable for the diagnosis of VKH as well as to assess the treatment response. The choroidal thickness can even exceed 800 µm during the active stage of the disease [7], which can make its evaluation using OCT difficult. Our evaluation of RPE and choroidal thickness has shown increased thickness of the RPE and total choroidal thickness during the active phase, which decreases post-resolution. Furthermore, there was a significant correlation between the RPE thickness and the total choroidal thickness during the active stage of the disease. Also, a previous study on normal RPE thickness in the general population has shown values that are similar to those obtained in the post-resolution phase of the disease in our study [14]. We propose that the RPE thickness can be used as a surrogate marker of choroidal inflammation during the active stage of the disease, which can be easily evaluated using OCT even when the choroidal imaging might be affected due to increased thickness and back shadowing.

We also saw during the examination of our cases that the overall reflectivity of the RPE, and especially the reflectivity of the RPE under the fovea, was significantly lower as compared with the reflectivity following the resolution of the disease. We hypothesize that during the active stage of the disease, fluid accumulation could be occurring inside the RPE cells, resulting in subclinical oedema and thereby decreasing the overall RPE reflectivity. This was again supported by the fact that we found RPE vacuolations in fifteen eyes, which might indicate a more severe intracellular fluid accumulation. We also kept in mind the possibility of decreased RPE reflectivity during the acute stage due to media haze and poorer imaging quality. However, on analysis of the image quality report obtained, there was no significant difference in the image quality score during the acute phase of the disease as well as following the resolution of the disease. We believe that the lower RPE reflectivity can also be considered a marker of the acute stage of VKH, especially when correlated with the thickness of the RPE in areas of the attached retina. The use of RPE reflectivity in choroidal disorders is not new. In a study by Maltsev et al., the authors showed an increased reflectivity of the RPE at sites of leakage in eyes with central serous chorioretinopathy, albeit using a different method to measure the RPE reflectivity [15]. Another morphological alteration that we analyzed was the subretinal deposits over RPE, both in the acute and resolved phases. These deposits could represent shed photoreceptors or photoreceptors in various stages of regeneration, as previously described by a study by Bae et al. [16] However, we could not demonstrate the restoration of the ellipsoid zone in these eyes due to poor documentation of subsequent follow-ups.

In terms of visual prognosis, we found that the duration of symptoms correlated with the baseline visual acuity. Furthermore, although all patients had good visual recovery, a worse baseline BCVA, a higher baseline CT, a lesser duration of symptoms, a higher baseline thickness of RPE under a detached retina, the presence of bacillary layer detachment, and the presence of hyperreflective choroidal dots were independently associated with greater improvement in visual acuity. The presence of these parameters could indicate a more severe form of the disease at baseline and a subsequent greater magnitude of improvement in visual acuity after resolution. However, on multivariable analysis, only the baseline BCVA and the presence of bacillary detachment at baseline, were seen to be significant. Agarwal et al. described the presence of the bacillary layer as a split in the photoreceptor inner segment myoid and suggested it to be an important finding in acute VKH [17]. We believe that this finding could also be related to the RPE changes seen in our study. The thickened/ inflamed RPE could be resulting in greater adhesion between RPE and photoreceptor, resulting in the fluid accumulation splitting the inner segment rather than the photoreceptor-RPE junction.

Some of the major strengths of this study include the good number of study eyes, the use of a single machine for image capture, and the utilization of good-quality images during the acute stage and follow-up, which adds to the validity of the results obtained. Furthermore, this is the first study that tried to examine the RPE in terms of thickening and reflectivity and how it correlated with the choroidal activity and its response to treatment. However, the retrospective nature of the disease is a limitation. Furthermore, we did not analyze RPE characteristics during recurrence, when the disease activity may be subclinical. Also, we did not confirm the resolution stage with the help of the FFA or ICGA. This could have resulted in some degree of uncertainty about whether the truly resolved phase of the disease has been analyzed. Third, the exclusion of eyes with poor imaging characteristics or a poorly defined outer choroidal boundary might have excluded severe forms of the disease.

## 5. Conclusions

In conclusion, we propose the utility of the RPE layer as a surrogate biomarker of choroidal activity and inflammation in terms of RPE reflectivity and thickness during the acute stage of VKH, which can be helpful, especially in cases where there is poor imaging of the choroid.

## Figures and Tables

**Figure 1 jcm-12-02368-f001:**
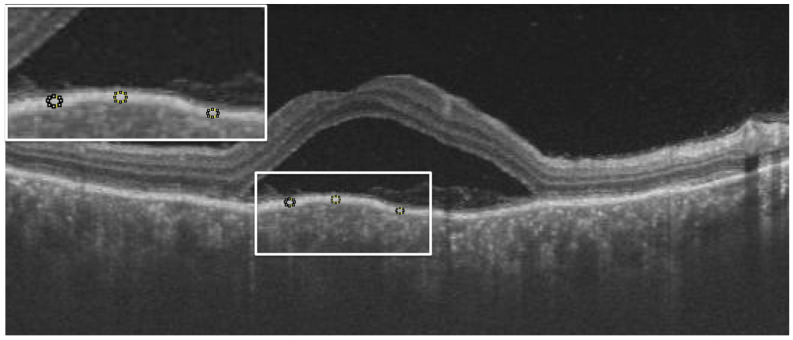
Example showing measurement of retinal pigment epithelial (RPE) reflectivity (white dotted circles in the white box inset at top left corner) below detached retina using ImageJ software. An average of the values from these three areas were taken for RPE reflectivity in detached retina. Similar measurements were taken under attached retina and subfoveally in all eyes.

**Figure 2 jcm-12-02368-f002:**
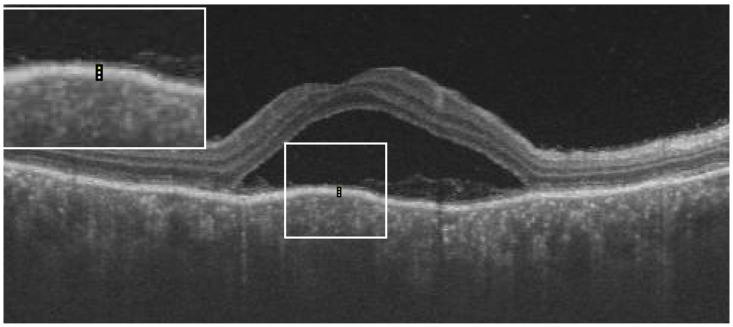
Example showing measurement of retinal pigment epithelial (RPE) thickness (white dotted line in the white box inset at the top left corner) using ImageJ software.

**Figure 3 jcm-12-02368-f003:**
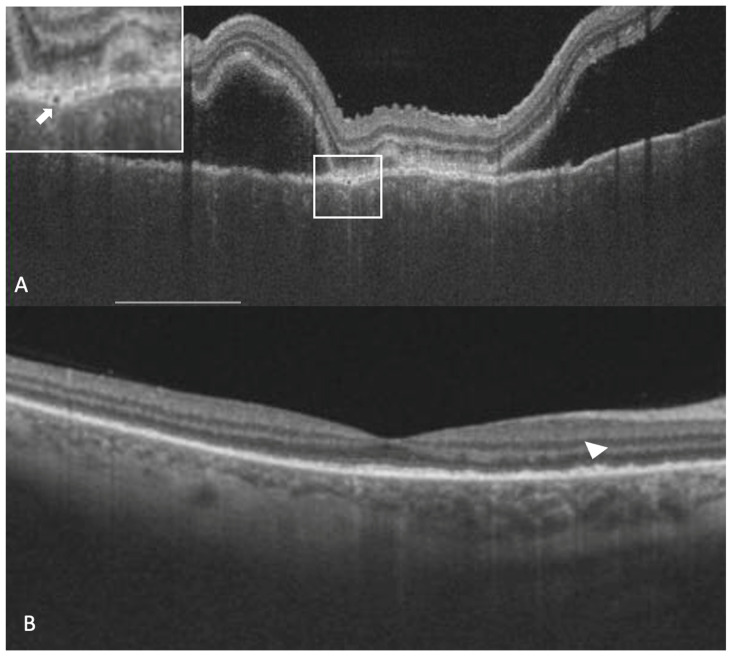
A 26-year-old lady with a diagnosis of acute Vogt-Koyanagi-Harada disease. The right eye shows sub-retinal fluid, thickened choroid, a thickened retinal pigment epithelium (RPE) and an RPE vacuolation (white arrow in the white box inset at the top left corner) in the vertical optical coherence tomography scan (**A**). After the resolution of sub-retinal fluid, persistent deposits over RPE could be noticed (arrowhead) (**B**).

**Figure 4 jcm-12-02368-f004:**
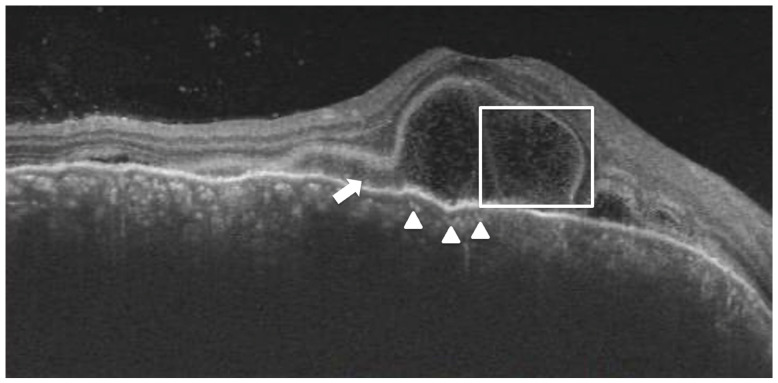
A 32-year-old lady with a diagnosis of acute Vogt-Koyanagi-Harada disease. The right eye shows bacillary layer detachment (white box) with subretinal hyperreflective material (white arrow) and choroidal hyperreflective dots (white arrowhead).

**Table 1 jcm-12-02368-t001:** Comparison of parameters during the acute stage of the disease and following resolution of disease activity.

Parameters	Baseline	Post Resolution	*p*-Value
BCVA (logMAR)	0.607 ± 0.447	0.110 ± 0.281	0.0001
Central macular thickness (microns)	616.396 ± 465.519	160.8 ± 24.980	0.0001 *
Overall RPE reflectivity	185.667 ± 10.304	189.948 ± 8.051	0.021
RPE reflectivity under fovea	189.297 ± 16.932	199.347 ± 19.310	0.001
RPE reflectivity under attached retina	192.546 ± 16.796	194.631 ± 17.949	0.507
RPE reflectivity under detached retina	173.938 ± 25.355	195.500 ± 11.881	0.0001
RPE thickness under fovea (microns)	32.288 ± 10.111	26.925 ± 5.555	0.0004
RPE thickness under attached retina (microns)	28.819 ± 8.682	21.565 ± 5.513	0.0001
RPE thickness under detached retina (microns)	32.801 ± 8.702	22.904 ± 5.530	0.0001
Sub foveal total choroidal thickness (microns)	589.831 ± 100.622	435.957 ± 115.009	0.0001 *

* Wilcoxon Signed Rank Test; BCVA: Best corrected visual acuity; logMAR: logarithm of minimum angle of resolution.

**Table 2 jcm-12-02368-t002:** Correlation between different parameters.

Parameters	Correlation Coefficient	Lower CI	Upper CI	*p*-Value
RPE reflectivity under attached retina post resolution versus	0.313	0.167	1.871	0.019
RPE thickness under attached retina post resolution				
RPE thickness under fovea pretreatment versus	0.373	0.011	0.063	0.004
Sub foveal total CT pre treatment				
Baseline BCVA versus	0.321	−0.015	−0.001	0.016
Duration of symptoms				

CT: Choroidal thickness; BCVA: Best corrected visual acuity.

**Table 3 jcm-12-02368-t003:** Linear regression analysis showing factors affecting change in visual acuity.

	Univariate			Multivariate		
	B	95% CI of B	*p*-Value	B	95% CI of B	*p*-Value
Baseline BCVA	−0.69	−0.84 to −0.54	<0.001	−0.63	−0.78 to −0.47	<0.001
Baseline total CT	−0.001	−0.002 to 0	0.026	−0.001	−0.001 to 0.001	0.92
Duration of symptoms	0.008	0.002 to 0.01	0.013	−0.001	−0.005 to 0.004	0.74
Baseline thickness of RPE under detached retina	−0.01	−0.03 to −0.002	0.024	−0.006	−0.014 to 0.002	0.15
Presence of bacillary detachment at baseline	−0.35	−0.54 to −0.16	0.001	−0.19	−0.34 to −0.05	0.01
Presence of hyper-reflective choroidal dots at baseline	−0.29	−0.59 to −0.004	0.047	0.04	−0.17 to 0.24	0.73

CT: Choroidal thickness; BCVA: Best corrected visual acuity.

## Data Availability

The data that support the findings of this study are available from the corresponding author, N.K.S., upon reasonable request.

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
