# Peer review of "Retinal Pigment Epithelial Characteristics in Acute and Resolved Vogt-Koyanagi-Harada Disease"

_jcm, 2023, doi:10.3390/jcm12062368_

Round 1
Reviewer 1 Report
This manuscript evaluated the role of RPE as biomarker for the VKH. The results are interesting and meaningful. The manuscript are well organized with clear writing. I only have several minor comments:
1. the measurement of the RPE thickness and the reflectivity can be well described with an image.
2. the RPE reflectivity should be well defined.
3. the connection between the RPE characteristics and the measurement of choroid can be conducted if it is possible.
Author Response
This manuscript evaluated the role of RPE as biomarker for the VKH. The results are interesting and meaningful. The manuscript are well organized with clear writing. I only have several minor comments:
Response: We thank the reviewer for the encouraging comments.
- The measurement of RPE thickness and reflectivity can be well described with an image
Response: We thank the reviewer for the suggestion. We have demonstrated how the RPE reflectivity and thickness was calculated with the help of an image as advised (Fig 4 and 5)
- RPE reflectivity should be well defined
Response: We thank the reviewer for pointing this out. “The oval selection tool of ImageJ was used to manually mark an elliptical region of interest within the RPE to measure the reflectivity in that area i.e. mean grey value.” (Page 2, para 4 Line 86). An image has also been added to demonstrate how the RPE reflectivity was calculated (Fig 4)
- The connection between RPE characteristics and the measurement of the choroid can be conducted if possible
Response: We thank the reviewer for pointing out this aspect. However, we could not find any association between the RPE morphological features like Bacillary layer detachment, focal RPE thickening, RPE vacuolations and choroidal thickness. “However, no correlation could be found between the above-mentioned RPE morphological features and the choroidal thickness.” (Page 4, para 2 Line 137)
Reviewer 2 Report
See attached document

Author Response
Reviewer 2:
- As the study background, “we speculated that few changes might be occurring at the level of the RPE that is causing these changes” (page 2, line 55). This speculation should be better substantiated with What is the importance of RPE in VKHD? There is very important literature in VKHD
Response: We acknowledge the concern raised by the reviewer and have gone through the articles suggested. We have added the relevant findings regarding the role of imaging RPE in VKH disease described in previous studies and mention why our study is novel with regards to RPE imaging (“While few imaging studies have analysed RPE, both following the resolution of the disease and in the chronic phase, describing features like focal areas of reversible RPE thickening and RPE atrophy, there is a lacuna in the literature regarding RPE imaging characteristics during the acute phase and immediately following resolution of the disease” in Page 2, para 2 Line 57)
- Vasconcelos-Santos DV, Sohn EH, Sadda S, Rao NA. Retinal pigment epithelial changes in chronic Vogt-Koyanagi-Harada disease: fundus autofluorescence and spectral domain-optical coherence tomography findings. Retina. 2010 Jan;30(1):33-41.
- Nakamura T, Hayashi A, Oiwake T. Long-term changes of retinal pigment epithelium in the eyes with Vogt-Koyanagi-Harada disease observed by adaptive optics imaging. Clin Ophthalmol. 2019 May 31;13:927-933.
- Miura M, Makita S, Azuma S, Yasuno Y, Sugiyama S, Mino T, Yamaguchi T, Agawa T, Iwasaki T, Usui Y, Rao NA, Goto H. Evaluation of Retinal Pigment Epithelium Layer Change in Vogt-Koyanagi-Harada Disease With Multicontrast Optical Coherence Tomography. Invest Ophthalmol Vis Sci. 2019 Aug 1;60(10):3352-3362.
- Since RPE thickness and reflectivity are not usual measurement parameters in VKHD nor in other ocular disorders or in normal eyes, should a sex and age-paired normal control group be considered. In this way, data could be bettered interpreted and appreciated. Previous papers demonstrated localized abnormalities
Response: We thank the reviewer for suggesting this modification. After a thorough literature search regarding RPE thickness, we found that the mean RPE thickness in the normal population is almost similar to the mean thickness of the RPE post-resolution of the VKH disease in our study. This has been added in the discussion at Page 9,Para 2, Line 209: “Also, a previous study on normal RPE thickness in the general population has shown values which are similar to what was obtained in the post-resolution phase of the disease in our study [14].”
- Ko F, Foster PJ, Strouthidis NG, Shweikh Y, Yang Q, Reisman CA, Muthy ZA, Chakravarthy U, Lotery AJ, Keane PA, Tufail A, Grossi CM, Patel PJ; UK Biobank Eye & Vision Consortium. Associations with Retinal Pigment Epithelium Thickness Measures in a Large Cohort: Results from the UK Biobank. Ophthalmology. 2017 Jan;124(1):105-117.
As we aimed at only demonstrating the changes in RPE characteristics post-resolution, we did not include normal controls. However, we will include it if the reviewer deems it to be necessary.
Also, to the best of our knowledge, there was only a single study in the literature regarding RPE reflectivity, described in a case of CSCR, where it was noted that there was a focal increase in the reflectivity of the RPE at sites of leakage. This has been added to discussion in Page 9, Para 3, Line 227: “The use of RPE reflectivity in choroidal disorders is not new. In a study by Maltsev et al, the authors showed an increased reflectivity of the RPE at sites of leakage in eyes with central serous chorioretinopathy, albeit using a different method to measure the RPE reflectivity [15].”
- Maltsev DS, Kulikov AN, Burnasheva MA, Kazak AA, Chhablani J. Retinal Pigment Epithelium Reflectivity at Leakage Site on Spectral-Domain Optical Coherence Tomography in Acute Central Serous Chorioretinopathy. Semin Ophthalmol. 2021 Aug 18;36(5-6):354-359.
- Concerning the population studied, include a brief treatment
Response: The treatment schedule of the study population has been added to the text as suggested (“All patients were treated with either oral steroids or intravenous steroids followed by oral steroids with gradual tapering of dose with most patients requiring additional immunomodulatory therapy”. Page 2, Line 78)
- The follow-up period should be mentioned
Response: The follow-up period has been mentioned.” The patients were followed up for a mean duration of 9.7+/-3.2 months.” Page 3, Para 4 Line 125)
- Points presented in results and discussion should be better explained in Materials and Methods, such as “improvement of visual acuity”.
Response: As per the comments, we have added a line in the methods: Page 2 para 3 line 73: “Improvement in visual acuity at resolution was defined as any decrease in the logarithm of minimum angle of resolution (logMAR) value from baseline.”
- Parameters presented in the results and discussion should be better defined, such as subretinal hyperreflective dots, choroidal hyperreflective dots, bacillary layer detachment, RPE vacuolations and focal RPE thickening (page 2 line 83,84). Special attention to RPE vacuolation that there is not much in the literature. Figures demonstrating each situation of the study should be presented in much detail in
Response: We apologize for the lack of clarity. “Subretinal hyperreflective dots were defined as hyperreflective dots below the neurosensory retina and over the RPE without associated back shadowing. Choroidal hyperreflective dots were defined as hyperreflective dots scattered throughout the choroidal stroma. Bacillary layer detachment was defined as a split of the neurosensory retina at the level of the myoid zone in the photoreceptor layer. RPE vacuolations were defined as hyporeflective cavities within the RPE and focal RPE thickening was defined as a focal hyperreflective lesion of the RPE”. (Page 3, para 2 Line 99)
We have also added OCT images describing the same (Fig 1, 2 and 3)
- The measurements were done at baseline and when “subretinal fluid completely
resolved” (Table 1) or when “post-resolution of disease” (page 2, line 72) or “post- treatment” (Page 2, line 83) or end of follow-up (Page 3, line 99)?
Response: We apologize for the lack of clarity. All the measurements were done at baseline and after resolution of VKH disease. This has been changed throughout the manuscript.
- Please explain the reflectivity measurement: pixel/area? density pixel/area?
Response: We thank the reviewer for pointing this out. ImageJ calculates the mean grey value of a selected area, which is an indirect measurement of reflectivity. This has been included in the methods: “The oval selection tool of ImageJ was used to manually mark an elliptical region of interest within the RPE to measure the reflectivity in that area i.e. mean grey value” (Page 2, Line 86). An image has also been added to demonstrate how the RPE reflectivity was calculated (Fig 4)
- Page 2, line 83: “Measurements post-treatment were done in locations identical to …“ insertion of the technology Smarttrack to make clear that the measurement was exactly in scans of the same
Response: We thank the reviewer for pointing this out. Image registration was done at baseline which enabled to obtain OCT images from the same location at subsequent visits (“Image registration was done for all follow-up imaging” in Page 2, para 3 Line 77; and “Measurements post-resolution of the disease were done in locations identical to the pretreatment measurements on the OCT B-scan, enabled by image registration.” In page 3 para 1 line 94)
- In statistical analysis, consideration for both eyes of the same patient should be
Response: The paragraph has been modified as suggested (“For analysis, both eyes were considered, if the disease was bilateral”. Page 3, Para 3, Line 106, “We performed a generalized estimating equation (GEE) to compensate the effect of both eyes of the same patient on statistical analysis.” Page 3, Para 3, Line 113)
- There were 29 patients and 55 eyes (page 3, line 95). Was there any unilateral case? VKHD mostly affects both eyes.
Response: Three eyes from three patients were excluded due to poor image quality or poor documentation. This has been included in Page 3, para 4, Line 119
- Page 3, line 97: “The mean duration of symptoms prior to presentation was 15.43 days and the mean duration of symptoms was 89.32 days”. What is the meaning of the “duration of symptoms”?
Response: We apologize for the lack of clarity. We have elaborated it in the text. “ The median duration of symptoms (decreased vision) prior to presentation to the clinic was 7 days (IQR 5 to 17.5 days) and the median duration till resolution of disease activity was 2.3 months (IQR 0.7 to 3.8 months). (Page 3, para 4 Line 120). The patients were followed up for a mean duration of 9.7+/-3.2 months. (Page 3, para 4 Line 125).
- In Table 1 “complete resolution of subretinal fluid”. When this occurred (mean, median, range).
Response:. Comparison of parameters was done between the acute stage of the disease and following the resolution of the disease activity. The legend in Table 1 has been corrected “Table 1: Comparison of parameters during acute stage of the disease and following resolution of disease activity” (Page 4, Line 153).
The median duration till resolution of disease activity was 2.3 months (IQR 0.7 to 3.8 months) (Page 3, Line 120).”The patients were followed up for a mean duration of 9.7+/-3.2 months.” (Page 3, Line 125). This has been corrected and is reflected in the methodology and results section.
- In Table 1, central macular thickness measurement at acute VKHD baseline with “177.6± 64.8 m” seems very odd
Response: We thank the reviewer for pointing out this mistake from our side. There was a measurement error while measuring the central macular thickness. We have calculated again and our values are as follows:
CMT ( Mean +/- SD; all values are in micrometers)
“Baseline
616.396 +/- 465.519
Final
160.8 +/- 24.980
Wilcoxon signed-rank test: p value < 0.0001” (Page 4, Table 1: Line 152)
- Also in Table 1, subfoveal choroidal thickness at baseline seemed relatively thin. Please check the reading.
Response: We acknowledge the concern raised by the reviewer. We have cross-checked the values again and found that there is no difference from our original results (Page 4, Table 1: Line 152). This could have been due to exclusion of eyes with poor image quality/ where outer choroidal boundary was not visible. This would have resulted in exclusion of very severe forms of disease with thicker choroids. This has been added in the limitations “Second, exclusion of eyes with poor imaging characteristics/ poorly defined outer choroidal boundary, might have excluded severe forms of the disease.” in page 10 para 2 line 254.
- Title: “Vogt Koyanagi Harada” disease should have hyphen between the names
Response: Apologies for the typographical error. We have corrected as pointed out in the title“Retinal Pigment Epithelial characteristics in acute and resolved Vogt-Koyanagi-Harada disease” and in other areas of the text
- All discussions should be reviewed with references adjusted to what is pointed
Response: The discussion of the manuscript has been modified based on the changes made.
- English should be
Response: Grammatical errors have been corrected.
Round 2
Reviewer 2 Report
Revised version: Retinal Pigment Epithelial characteristics in acute and resolved Vogt-Koyanagi-Harada disease
The authors tried to answer each query. However, there are still many major concerns about this paper.
While examining the pictures describing RPE thickening/increase in reflectivity, one could argue that in fact these findings represent photoreceptors under recovery (M Zhou et al Eye 2018; 32:572–578). The nomenclature of subretinal material in VKHD has been continuously being reviewed (Bae SS et al 2016). In Figure, “persistent deposits over RPE” seems the same as in Figure 3 “focal areas of RPE thickening”. What the authors think?
There are conceptual misunderstandings:
- VKH disease is a bilateral disease, so it is not adequate to write “…, if the disease was bilateral.” (P 3 l 106)
- “choroidal hyperreflective dots… throughout the choroidal stroma.” (P3 l 100) how about the choriocapillaris?
- “… focal RPE thickening was defined as a focal hyperreflective lesion of the RPE” (p 3 l 104) The authors mention RPE thickness AND RPE reflectivity as separate outcomes!
- “resolution of disease” (many parts, i.e. p2 l 74) what the authors mean by resolution? Inflammatory signs such as cells in anterior chamber, serous retinal detachment, fluorescein angiographic signs, indocyanine green angiographic signs, OCT signs… How the authors characterized resolution of disease?
Author Response
Comment 1: The authors tried to answer each query. However, there are still many major concerns about this paper.
While examining the pictures describing RPE thickening/increase in reflectivity, one could argue that in fact these findings represent photoreceptors under recovery (M Zhou et al Eye 2018; 32:572–578). The nomenclature of subretinal material in VKHD has been continuously being reviewed (Bae SS et al 2016). In Figure, “persistent deposits over RPE” seems the same as in Figure 3 “focal areas of RPE thickening”. What the authors think?
Response: We thank the reviewer for suggesting the interesting articles. After going through the articles, we believe the focal RPE thickenings/ RPE deposits described in our series could represent the shed photoreceptors/ photoreceptors under recovery captured during various stages of photoreceptor recovery. We have deleted the term RPE thickening in order to avoid confusion and added the explanation for the RPE deposits in the discussion. As our documentation of subsequent follow-ups were not adequate due to the retrospective nature of the study, we could have missed the complete photoreceptor regeneration stage. We have also removed Figure 3 to avoid confusion.
Page 9 para 2 line 229: “Another morphological alteration that we analysed were the subretinal deposits over RPE, both in the … subsequent follow-ups.”
Comment 2: There are conceptual misunderstandings:
- VKH disease is a bilateral disease, so it is not adequate to write “…, if the disease was bilateral.” (P 3 l 106)
Response: We apologize for this error. We have rephrased our sentence to “For analysis, both eyes were considered. Eyes were excluded from analysis in case of poor image quality or poor documentation.” (Page 3: Line 110-111)
Comment 3: - “choroidal hyperreflective dots… throughout the choroidal stroma.” (P3 l 100) how about the choriocapillaris?
Response: We thank the reviewers for raising this query. The choroidal hyperreflective dots which we have described in our paper were defined as hyperreflective dots scattered throughout the choroid. We have rephrased the sentence to “Choroidal hyperreflective dots were defined as circumscribed dots within choroid, seen on OCT scan having equal or higher reflectivity than RPE band.” (page 3 para 2 line 103)
Comment 4: - “… focal RPE thickening was defined as a focal hyperreflective lesion of the RPE” (p 3 l 104) The authors mention RPE thickness AND RPE reflectivity as separate outcomes!
Response: We thank the reviewers for pointing this out. The term RPE thickening has been removed to avoid confusion as per previous comments and we have only included deposits over RPE for analysis.
Comment 5: “resolution of disease” (many parts, i.e. p2 l 74) what the authors mean by resolution? Inflammatory signs such as cells in anterior chamber, serous retinal detachment, fluorescein angiographic signs, indocyanine green angiographic signs, OCT signs… How the authors characterized resolution of disease?
Response: We apologize for the lack of clarity. We have added a definition of resolution: “Resolution of disease was determined as per the discretion of the treating ophthalmologist based on the resolution of inflammatory signs such as cells in anterior chamber and complete resolution of subretinal fluid on OCT.” (Page 2, para 3, Line 74) We could not confirm the resolution on FFA or ICG documentation due to the retrospective nature of the study. This has been added to the limitation. “Also, we did not confirm the resolution stage with the help of FFA or ICGA. This could have resulted in some degree of uncertainty whether the truly resolved phase of the disease has been analysed.” (Page 10, para 1, line 254)
All the figures have been rearranged and cited in the text.
